# Deep Intronic ETFDH Variants Represent a Recurrent Pathogenic Event in Multiple Acyl-CoA Dehydrogenase Deficiency

**DOI:** 10.3390/ijms25179637

**Published:** 2024-09-05

**Authors:** Stefania Martino, Pietro D’Addabbo, Antonella Turchiano, Francesca Clementina Radio, Alessandro Bruselles, Viviana Cordeddu, Cecilia Mancini, Alessandro Stella, Nicola Laforgia, Donatella Capodiferro, Simonetta Simonetti, Rosanna Bagnulo, Orazio Palumbo, Flaviana Marzano, Ornella Tabaku, Antonella Garganese, Michele Stasi, Marco Tartaglia, Graziano Pesole, Nicoletta Resta

**Affiliations:** 1Medical Genetics Unit, Department of Precision and Regenerative Medicine and Ionian Area (DiMePRe-J), University of Bari “Aldo Moro”, 70124 Bari, Italy; s.martino5@studenti.uniba.it (S.M.); antotur90@gmail.com (A.T.); alessandro.stella@uniba.it (A.S.); rosanna.bagnulo@uniba.it (R.B.); nelatabaku96@gmail.com (O.T.); michele.stasidoc@libero.it (M.S.); 2Department of Biosciences, Biotechnologies & Environment, University of Bari “Aldo Moro”, Via Edoardo Orabona 4, 70125 Bari, Italy; pietro.daddabbo@uniba.it (P.D.); graziano.pesole@uniba.it (G.P.); 3Molecular Genetics and Functional Genomics, Ospedale Pediatrico Bambino Gesù, IRCCS, Viale di San Paolo 15, 00146 Rome, Italy; fclementina.radio@opbg.net (F.C.R.); cecilia.mancini@opbg.net (C.M.); marco.tartaglia@opbg.net (M.T.); 4Department of Oncology and Molecular Medicine, Istituto Superiore di Sanità, Viale Regina Elena 299, 00161 Rome, Italy; alessandro.bruselles@iss.it (A.B.); viviana.cordeddu@iss.it (V.C.); 5Section of Neonatology and Neonatal Intensive Care Unit, Department of Interdisciplinary Medicine, University of Bari “Aldo Moro”, 70121 Bari, Italy; nicola.laforgia@uniba.it (N.L.); dottcapodiferro@virgilio.it (D.C.); 6Clinical Pathology and Neonatal Screening, Hospital “Giovanni XXIII”, University Hospital Consortium Corporation Polyclinics of Bari, 70124 Bari, Italy; simonetta.simonetti@policlinico.bari.it; 7Division of Medical Genetics, Fondazione IRCCS—Casa Sollievo della Sofferenza, San Giovanni Rotondo, 71013 Foggia, Italy; o.palumbo@operapadrepio.it; 8Institute of Biomembranes, Bioenergetics and Molecular Biotechnologies, Consiglio Nazionale delle Ricerche, Via Amendola 122/O, 70126 Bari, Italy; f.marzano@ibiom.cnr.it; 9Medical Genetic Unit, University Hospital Consortium Corporation Polyclinics of Bari, 70124 Bari, Italy; a.garganese80@gmail.com

**Keywords:** *ETFDH*, deep intronic variant, MADD, genome sequencing, RNA sequencing, pseudo-exon, transcript processing, splicing

## Abstract

Multiple acyl-CoA dehydrogenase deficiency (MADD) is a rare inborn error of metabolism affecting fatty acid and amino acid oxidation with an incidence of 1 in 200,000 live births. MADD has three clinical phenotypes: severe neonatal-onset with or without congenital anomalies, and a milder late-onset form. Clinical diagnosis is supported by urinary organic acid and blood acylcarnitine analysis using tandem mass spectrometry in newborn screening programs. MADD is an autosomal recessive trait caused by biallelic mutations in the *ETFA*, *ETFB*, and *ETFDH* genes encoding the alpha and beta subunits of the electron transfer flavoprotein (ETF) and ETF-coenzyme Q oxidoreductase enzymes. Despite significant advancements in sequencing techniques, many patients remain undiagnosed, impacting their access to clinical care and genetic counseling. In this report, we achieved a definitive molecular diagnosis in a newborn by combining whole-genome sequencing (WGS) with RNA sequencing (RNA-seq). Whole-exome sequencing and next-generation gene panels fail to detect variants, possibly affecting splicing, in deep intronic regions. Here, we report a unique deep intronic mutation in intron 1 of the *ETFDH* gene, c.35-959A>G, in a patient with early-onset lethal MADD, resulting in pseudo-exon inclusion. The identified variant is the third mutation reported in this region, highlighting *ETFDH* intron 1 vulnerability. It cannot be excluded that these intronic sequence features may be more common in other genes than is currently believed. This study highlights the importance of incorporating RNA analysis into genome-wide testing to reveal the functional consequences of intronic mutations.

## 1. Introduction

Multiple acyl-CoA dehydrogenase deficiency (MADD, MIM 231680), also known as glutaric acidemia II, is a rare recessive, genetically heterogeneous, combined disorder of fatty acid and amino acid oxidation, affecting approximately 1 in 200,000 live births with ethnic variations. Three clinical phenotypes with differences in presentation and age of onset have been recognized, including two MADD-severe (MADD-S) neonatal-onset forms, with or without congenital anomalies, and a MADD-mild (MADD-M) late-onset form [1]. MADD-S generally presents with non-ketotic hypoglycemia, hypotonia, hepatomegaly, and severe metabolic acidosis within the first 24 h of life, evolving into death early after birth. The associated congenital anomalies usually include dysplastic kidneys with multiple cysts; facial dysmorphism (e.g., low-set ears, high forehead, hypertelorism and hypoplastic midface); rocker-bottom feet; and anomalies of external genitalia. MADD-S without congenital anomalies generally has a neonatal onset with hypotonia, tachypnea, hepatomegaly, metabolic acidosis, and hypoketotic hypoglycemia. The majority of individuals die soon after onset. Those who survive show an evolutive cardiomyopathy. MADD-M has a broader clinical spectrum, ranging from intermittent episodes of vomiting, metabolic acidosis, and hypoketotic hypoglycemia with or without cardiac involvement in infancy to acute ketoacidosis and lipid storage myopathy in adolescents/adults. In all forms, urinary organic acid analysis typically reveals various combinations of increased dicarboxylic acids, glutaric acid, ethylmalonic acid, 2-hydroxyglutarate, and glycine conjugates. Blood acylcarnitines show increased C4-C18 species, although patients may be severely carnitine depleted, which can limit the extent of these abnormalities.

MADD can be screened using tandem mass spectrometry, which is an informative tool used in newborn screening programs. The identification of abnormalities, in particular in acylcarnitine levels, may also confirm the diagnosis, and plasma acylcarnitine profiling may suggest a block in fatty acid oxidation before symptoms appear [2]. First-line evaluation is generally provided by newborn screening, the positive result of which is confirmed by second-generation sequencing approaches. MADD is caused by pathogenic variants in the *ETFA* (MIM *608053; glutaric acidemia IIA), *ETFB* (MIM *130410; glutaric acidemia IIB), and *ETFDH* (MIM *231675; glutaric acidemia IIC) genes, which encode the alpha and beta subunits of the electron transfer flavoprotein (ETF) and ETF-coenzyme Q oxidoreductase [3,4].

To date, more than 900 *ETFDH* variants have been reported in the ClinVar database (last accessed 13 May 2024), including nearly 320 variants that have been classified as either pathogenic or likely pathogenic. Given that the condition is inherited as an autosomal recessive trait, establishing the molecular diagnosis requires the identification of biallelic disease-causing variants. MADD-S is caused by loss-of-function (LoF) variants, including those resulting in truncated forms of the encoded protein or causing aberrant mRNA expression, processing, or stability. On the other hand, missense mutations are generally linked to the milder late-onset presentation of MADD [5].

The implementation of novel high-throughput technologies, such as second-generation sequencing, has greatly improved the diagnostic yield of gene testing in MADD. The present report investigates a patient clinically diagnosed with early-onset MADD where clinical exome sequencing (CES) revealed a single likely pathogenic variant in *ETFDH* predicting a stop codon loss. Subsequently, chromosome microarray analysis (CMA) and whole-genome sequencing (WGS) analysis were carried out. CMA resulted negative for the presence of copy number variations (CNVs) involving the *ETFDH* gene, while WGS disclosed the presence of a previously unreported deep intronic variant that was predicted to affect transcript processing within a region in which pathogenic MADD variants had previously been reported [6,7]. In silico and functional validation analyses confirmed the clinical relevance of the identified intronic variant, highlighting the significance of RNA-based investigations in obtaining a conclusive genetic diagnosis and allowing correct genetic counseling and patient care.

## 2. Materials and Methods

### 2.1. Patient

The patient was born at 32 weeks of gestation by urgent caesarean section due to cardiotocographic alterations in a primigravida with a pregnancy complicated by oligohydramnios and intrauterine fetal growth retardation from the 31st week, significantly hypotonic. Family history was negative, and parents were not related. Birth weight was 1470 g (24th centile), length 40 cm (21st centile), head circumference 31 cm (84th centile). Apgar 9-9. Non-invasive ventilation with n-CPAP and parenteral nutrition via umbilical venous catheter were applied. From the third day of life, the progressive deterioration of her clinical condition with hyporeactivity prompted the need for mechanical ventilation. Hyperammonemia (450 mmol/L; n.v < 100) and metabolic acidosis with increased anion gap (27 mmol/L) were observed; parenteral solutions were substituted with only 10% glucose; and treatment with carglumic acid, bioarginine, and sodium bicarbonate was started. She was transferred on the fourth day of life to our NICU, where elevated levels of proline (736 mmol/L; upper limit 254 mmol/L), valine (492.3 mmol/L; upper limit 217 mmol/L), aminoisobutyrate (231 mmol/L; upper limit 24 mmol/L), ornithine (593.5 mmol/L; upper limit 129 mmol/L), and lysine (511.6 mmol/L; upper limit 200) were found, both in plasma and urine. Additionally, both quantitative and qualitative organic acid analysis by gas chromatography-mass spectrometry (GC/MS) were performed, revealing a significant increase in glutaric, phenylacetic, adipic, and lactic acids, as well as the presence of isovalerylglycine and butyrylglycine in urine. Intravenous carnitine was added to the previous treatment. Despite the reduction in ammonemia with a slight improvement of metabolic acidosis, the patient’s clinical condition, with evident deep comatose state, did not improve. Elevated transaminases (GOT 1241 U/L, n.v < 34; GPT 227 U/L, n.v < 49), CPK 3301 U/L, n.v < 303; and LDH 5445 U/L, n.v < 629, were also found. Severe hypoxemia and hypotension were unresponsive to oscillatory ventilation as well as high doses of intravenous dopamine and dobutamine, leading to her death on the eighth day of life.

### 2.2. Molecular Analyses

Genomic DNA (gDNA) was extracted from proband and parents’ peripheral blood (PB) samples by using a QIAamp Mini Kit (Qiagen, Hilden, Germany), following the manufacturer’s instructions. Total RNA was extracted from PB specimens taken from both parents and unaffected sex- and age-matched individuals, collected in PAXgene Blood RNA tubes (PreAnalytiX, Qiagen), and purified by a PAXgene Blood RNA kit (PreAnalytiX, Qiagen) according to the manufacturer’s indications. gDNA and RNA were quantified with a BioSpectrometer Plus instrument (Eppendorf, Hamburg, Germany).

Trio-based CES was performed using the TruSight One Sequencing Panel kit (Illumina, San Diego, CA, USA). gDNA concentration and quality were evaluated by using a Qubit dsDNA HS Assay Kit on a Qubit 2.0 Fluorimeter (Invitrogen, Carlsbad, CA, USA), following the manufacturer’s instructions. Libraries were prepared by utilizing the NextEra Flex for Enrichment protocol (Illumina) and were sequenced on the NextSeq550Dx Illumina platform (Illumina).

Data analysis was performed by using the NextSeq control software v.4.2.0 and Local Run Manager software v4.0.0, both provided by Illumina (Illumina). Reads were aligned against the human genome reference (GRCh38) by the BWA Aligner software v.11.5 [8]. Variant calling was performed using the Genome Analysis Toolkit (GATK) [9]. A mean coverage depth of 229× was obtained. Variant calling data were analyzed with Geneyx analysis software v. 5.12 (Geneyx, Herzliya, Israel). Variants were filtered and prioritized by utilizing HPO terms (hyperammonemia, glutaric aciduria, and hyperlactacidemia) [10], using in silico tools (Alamut Visual Plus, MaxEntScan, SpliceAI) and public databases (ClinVar, LOVD, Varsome, Franklin by genoox, OMIM), and were classified following the ACMG criteria [11]. BAM files were visually inspected by the Integrative Genome Viewer software 2.16.0 (IGV) and Alamut Visual Plus Genome Viewer (Sophia Genetics, Lausanne, Switzerland), and variants were reported according to the Human Genome Variants Society (HGVS) recommendations [12].

CMA was performed using the CytoScan XON array (Thermo Fisher Scientific, Waltham, MA, USA). The CytoScan XON assay was performed according to the manufacturer’s protocol, using 100 ng of DNA. XON array data were analyzed for the presence of intragenic microdeletions/duplications using the Affymetrix Chromosomal Analysis Suite (ChAS) software v.4.3. CNV pathogenicity was assessed using the published literature and public databases (Database of Genomic Variant, http://dgv.tcag.ca/dgv/app/home, accessed on 15 July 2024; Clingen, https://clinicalgenome.org/, accessed on 15 July 2024; DECIPHER, https://www.deciphergenomics.org/, accessed on 15 July 2024; OMIM, https://www.omim.org/, accessed on 15 July 2024). Genomic positions, functional annotation on genomic regions, and genes affected by CNVs and/or ROHs were derived from the University of California Santa Cruz Genome Browser tracks (http://genome.ucsc.edu/cgi-bin/hgGateway, accessed on 15 July 2024). The clinical significance of each rearrangement was assessed according to the ACMG and Clinical Genome Resource 2020 guidelines [13].

WGS was performed on a NovaSeq 6000 platform (Illumina) as per recommended protocols. Base calling and data analysis were performed using Bcl2FASTQ (Illumina). Paired-end read mapping to the GRCh38 reference sequence, variant calling, and joint genotyping were run using Sentieon v.2023-08 (https://www.sentieon.com, accessed on 17 July 2024). SNP and short insertion/deletion (InDels) hard filtering were applied using GATK, version 3.8.0 (Broad Institute, Cambridge, England). High-quality variants were first filtered by frequency ≤ 5% in the in-house WGS population-matched database. Variants were annotated and filtered against public (gnomAD v.2.1.1, https://gnomad.broadinstitute.org, accessed on 17 July 2024) and in-house (>3100 population-matched exomes) databases to retain private and rare (MAF < 0.1%) variants with any effect on the coding sequence, and within splice site regions. The predicted functional impact of variants was analyzed by Combined Annotation Dependent Depletion (CADD) v.1.6 [14], M-CAP v.1.3 [15] and InterVar v.2.2.2 algorithms [16]. Clinical interpretation followed the ACMG 2015 guidelines [11]. Variants within non-coding regions were annotated and prioritized using Genomiser [17] (phenotype data version 2302).

Structural variants were detected using DELLY v.1.1.6 [18] and prioritized using AnnotSV v.3.3.2 [19]. WGS metrics and sequencing output are reported in Appendix A.

RNA-seq was performed using a NovaSeq6000 platform (Illumina). Raw sequences were inspected using FASTQC [20] and trimmed using FASTP [21]. High-quality reads were aligned onto the GRCh38 assembly of the human genome, using STAR 2.7.11a and providing a list of known gene annotations from GENCODE v43. Detected splicing junctions were classified based on their annotation status in the human transcriptomes. Data were also graphically inspected using IGV and custom tracks on the UCSC genome browser. High-quality reads were analyzed by Salmon v1.10.0, also providing the GENCODE v43 annotations, to estimate transcript level abundance from the RNA-seq data.

All the clinically relevant variants identified by CES, WGS, and RNA-seq were confirmed by Sanger sequencing (Applied Biosystem, Waltham, MA, USA). To validate the identified aberrant transcripts, a targeted cDNA assay was carried out. cDNA was synthetized by using the Thermo Scientific Maxima Reverse Transcriptase kit (Thermo Fisher Scientific), following the manufacturer’s recommendations. cDNA PCRs were performed as previously described [7]. Primer sequences are available upon request. The obtained PCR amplicons were separated via agarose gel electrophoresis to visually inspect the different amplification patterns. Appendix A lists the specific primers used to detect RNA isoforms (designed according to RNA-seq data) and to validate the presence of disease-causing variants both in exon 13 and intron 1.

The predicted functional consequences of the identified deep intronic variants were inspected using the splicing module of Alamut Visual Plus version 1.6.1, (Sophia Genetics). Splicing donor site (SDS) and splicing acceptor site (SAS) scores were calculated via the Maximum entropy (MaxEnt) of the Burge Laboratory’s MaxEntScan web-tool [22]. Finally, SpliceAI (https://spliceailookup.broadinstitute.org, accessed on 17 July 2024) was used to explore the gain and/or loss of splicing sites [23].

## 3. Results

Trio-based CES revealed the heterozygous and paternally inherited variant c.1852T>C (p.*618Glnext*13) in the *ETFDH* gene (NM_004453.4), which was subsequently validated by Sanger sequencing (Figure 1). The variant was classified as likely pathogenic according to the ACMG criteria (PP5, PM2, PM4) and to the LOVD database (https://www.lovd.nl/, accessed on 15 July 2024).

Since the clinical features of the proband well fitted the recessive condition associated with LoF variants in *ETFDH*, complementary approaches were performed to identify a possible second disease-causing variant in trans. Since XON testing disclosed no CNVs involving the *ETFDH* gene, trio-based WGS analysis was carried out, identifying a deep intronic variant (c.35-959A>G) within the *ETFDH* (NM_004453.4) intron 1. Sanger sequencing validated the variant, documenting its maternal transmission (Figure 1).

At the top, the Alamut Visual Plus (V 1.8.1) screenshot of the first three exons and last four exons of the ETFDH gene (NM_004453.4) is depicted. The deep intronic variant found in the patient is in the first intron. The paternally inherited mutation occurred in exon 13, the last of the ETFDH (NM_004453.4) gene. For each variant, the electropherograms results are displayed. The left panel shows the intronic variant, indicated by the red arrows, both in the proband and in her mother. In the right panel, the presence of the exonic variant, detected in the proband and in the father’s PB specimen, is shown, indicated by the blue arrows.

Then, we investigated the potential impact of the deep intronic variant on splicing using a variety of in silico prediction tools. Notably, two other deep intronic variants, c.35-768A>G and c.35-1008T>G, which are located in the same intronic region, were previously shown to cause abnormal mRNA processing [7] by including a cryptic pseudo-exon in the final mRNA coding sequence. Therefore, we first verified the creation/abrogation/modification of Exonic Splicing Enhancer (ESE) sites with the splicing module of Alamut Visual. Similarly to what was observed for the two previously reported intronic variants, the c.35-959A>G variant was predicted to create a novel serine/arginine-rich (SR) protein splicing factor 2 (SF2/ASF) ESE and perturb the strength of an SF2/ASF IgM-BRCA1 ESE (the prediction score increased from 2.38 to 4.16) (Figure 2a). Next, we used MaxEnt scan to assess if novel splicing donor or acceptor sites were generated or abolished. The three tested variants similarly created novel splice donor sites (SDSs), which outscored the wild-type SDS of intron 1 in all cases (Figure 2b). The pseudo-exon generated by the three variants was predicted to share a common SAS, which again outscored the wild-type SAS at the end of intron 1. Finally, wild-type and mutated sequences were inspected with Splice AI. Once more, for all three variants (c.35-768 A>G; c.35-959 A>G; c.35-1008 T>G), a significant increase in the delta score for the donor site of pseudo-exons (≥0.80) was observed [24] (Figure 2c). Overall, these in silico predictions consistently indicated a putative impact of the identified deep intronic variant on transcript processing.

To validate the in silico predictions, transcriptomic analyses were conducted using mRNA from parental PB samples. Consistently, splicing junction detections by STAR allowed the identification of an aberrant processing of the maternal *ETFDH* transcript, which was not present in the GENCODE v43 database. The variant *ETFDH* mRNA included an intronic sequence downstream of the first exon that was retained due to the use of a cryptic SAS located in the first intron of the gene (Figure 3a). The introduced sequence resulted in a shift of the reading frame and premature termination of the encoded protein (Figure 3a).

Targeted RNA analysis was conducted using maternal PB-derived cDNA to confirm this finding, documenting heterozygosity for the wild-type isoform (an expected amplicon size of 206 bp) and an aberrant amplicon (an amplicon size of 424 nucleotides) (Figure 3b), the latter resulting from the insertion of the 218 nucleotide-long retained intronic region between the canonical exons 1 and 2 (Figure 3b,c).

## 4. Discussion

Despite the remarkable progress in second-generation sequencing techniques, a significant number of patients still lack a definitive molecular diagnosis, which inevitably impacts their access to clinical care and proper genetic counseling [7]. Here, by coupling WGS to RNA-seq analysis, we reached a definitive molecular diagnosis for a rare disease compatible with the clinical symptoms occurring in the affected newborn, for which CES had allowed only the identification of a single pathogenic variant associated with the suspected condition.

In fact, methods based on whole-exome sequencing or next-generation gene panels do not allow for identifying variants in deep intronic regions. Thus, it is crucial to employ combined approaches that include patients’ RNA sequencing analysis to identify the plethora of aberrations affecting the splicing process. Mutations within intronic regions can disrupt this finely tuned mechanism, resulting in the inclusion of pseudo-exons, segments of intronic DNA erroneously recognized as exons, in mRNA by the splicing machinery [7]. To date, numerous cases of deep intronic mutations have been identified across various genetic diseases that, with distinct pathogenetic mechanisms, ultimately lead to altered gene expression [25]. Besides the inclusion of pseudo-exons, the ablation of transcriptional regulatory motifs, genomic rearrangements, the activation of cryptic splice sites, and the inactivation of intron-encoded RNA genes represent common events. The inclusion of pseudo-exons is usually due to mutations that create novel splice donor or acceptor sites, or activate existing cryptic splice sites [26]. Numerous studies have reported genomic variants that lead to the inclusion of pseudo-exons, including those identified in the *ETFDH* gene [7,27]. Given the number of splicing-altering variations reported in this region, the identification of a third unique deep intronic mutation resulting in a pseudo-exon inclusion in the *ETFDH* gene in a patient with early-onset MADD emphasizes the natural vulnerability of intron 1 in the *ETFDH* gene. Generally, the inclusion of pseudo-exons in the mature mRNA transcript can lead to frameshifts, premature stop codons, or to the insertion of noncanonical amino acid sequences, generally generating nonfunctional proteins or proteins with a deleterious function. The deep intronic heterozygous variant c.35-959A>G described in this work is the third mutation reported in intron 1 of the *ETFDH* gene that leads to a pseudo-exon activation [7]. Therefore, the variant was classified as likely pathogenic according to the ACMG criteria (PS3, PM2). We demonstrate that these variants create novel exonic splicing enhancers (ESEs), leading to pseudo-exon inclusion through the formation of a new splicing donor site that successfully competes with the natural donor site, altering normal splicing. The creation of a novel SDS is the commonest mechanism of pseudo-exon activation in the human genome [26]. Indeed, the three variants determined an increase in the SD score from approximately 0 to well above the score of the natural SD site computed by the MaxEntScan (MES) algorithm. Further, both the c.35-959A>G herein described and c.35-1008T>G activate a pseudo-exon whose size is in the range of those reviewed in Vaz-Drago et al. [25], while variant c.35-768A>G generates a 410 bp pseudo-exon that would represent one of the largest reported so far. The identification of three deep intronic variants in a stretch of 240 nucleotides is rather unusual. The *ETFDH* intron 1 pseudo-exon, even in the wild-type sequence, possesses a strong acceptor site at its 5′ end, as evidenced by its MES score that is higher than the normal acceptor in *ETFDH* exon 2. Thus, it is possible that any nucleotide variation, downstream of the pseudo-exon acceptor creating a strong novel donor site, could promote the rescue of a pseudo-exon. The identification of a mutation that promotes pseudo-exon inclusion highlights the necessity of extending genetic testing beyond conventional exonic mutations, particularly when the coding sequence screening has been negative. Also, our findings underscore the potential for intronic and splicing-related mutations to contribute to the phenotypic variability observed in MADD. Standard DNA sequencing methods may overlook deep intronic mutations and their consequent impact on splicing. Therefore, integrating RNA analysis into 343 genome-wide testing can provide valuable insights into the functional consequences of 344 intronic mutations, revealing aberrant splicing events that contribute to disease pathology [28]. This approach not only enables a more accurate genetic diagnosis, effective genetic counseling, and improved patient management, but also paves the way for future therapeutic interventions targeting splicing mechanisms. These interventions could potentially modulate splicing patterns and restore normal gene function. However, for MADD, there is currently no evidence supporting this possibility.

## Figures and Tables

**Figure 1 ijms-25-09637-f001:**
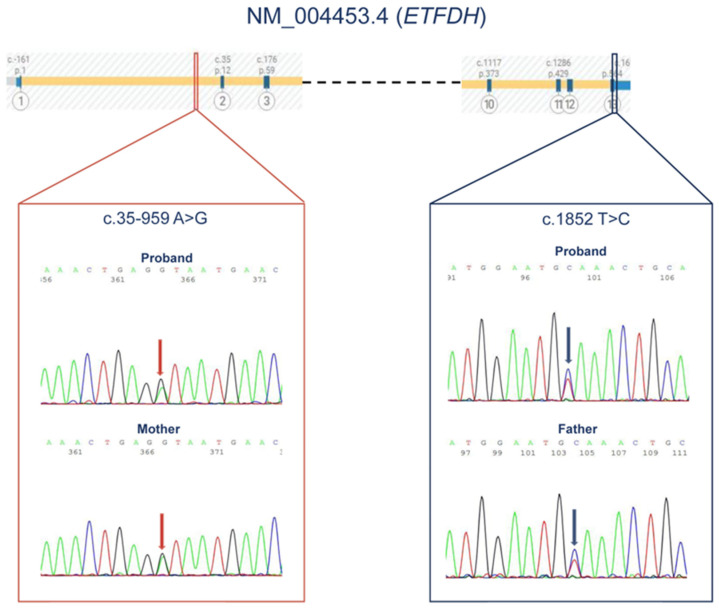
Sanger sequencing results of confirmatory and segregation analysis.

**Figure 2 ijms-25-09637-f002:**
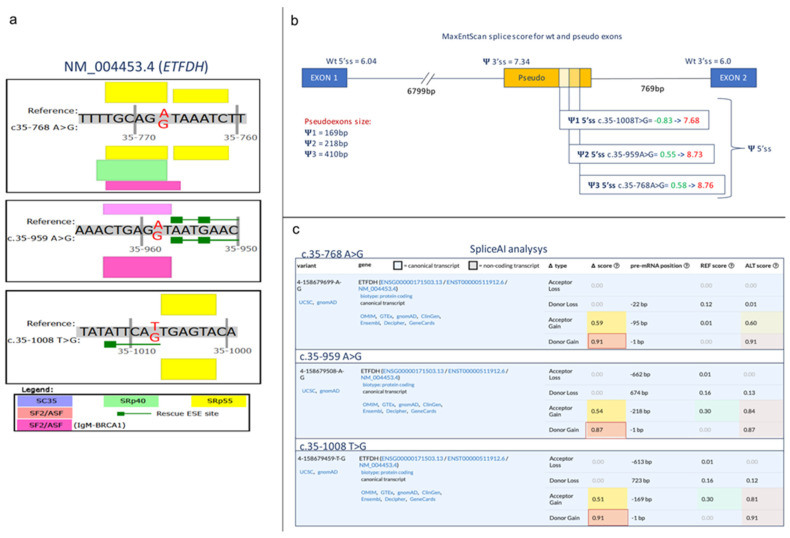
In silico prediction of the potential impact of the reported deep intronic variant on splicing. (**a**) Outline of ESE changes determined by the three deep *ETFDH* intron 1 mutations. Box height is proportional to ESE score as computed by the Alamut splicing module. (**b**) MaxEntScan analysis of 5′ and 3′ splicing site scores for both wild-type exon 1 and exon 2 junctions and the pseudo-exon. In green is the score of the normal sequence; in red, the score of the mutated sequence for the pseudo-exon. (**c**) SpliceAI analysis. The scores for the wild-type and mutated sequences are indicated for all three mutations in *ETFDH* intron 1. Boxed in red is the difference (i.e., ∆ score) between normal and mutated sequence scores for the donor site. The light red color indicates that the increase in SpliceAI scores for donor sites is highly significant (for all three mutations > 0.80) as reported by Sagakuchi et al. [24].

**Figure 3 ijms-25-09637-f003:**
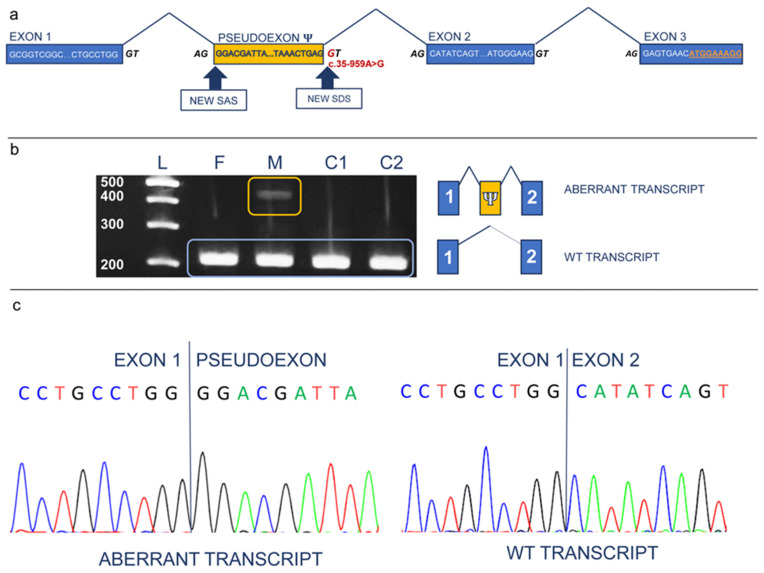
Confirmatory targeted RNA analysis on maternal PB-derived cDNA. (**a**) The blue rectangles represent the first three exons of the *ETFDH* gene present in both the normal and the aberrant transcript. In yellow, the pseudo-exon included in the abnormal mRNA. For each exon, and for the pseudo-exon, the sequence of first and last nucleotides is indicated. The splicing donor and acceptor sites are also depicted. The additional SAS and SDS created by the intronic mutation (in red) are indicated by the arrows. The aberrant isoform has a start codon within exon 3, with the novel coding sequence in orange, underlined. (**b**) RT-PCR experiment on *ETFDH* cDNA. On the left, the gel electrophoresis image is shown. In the first lane, the ladder (L) is present with 200, 300, 400, and 500 bp long DNA fragments. The second (F) and third (M) lane are the PCR products from the father’s and mother’s cDNA, respectively. The fourth and fifth lane (C1, C2) indicate the negative control PCR products. WT isoforms are boxed in light blue, while the novel aberrant transcript is present only in the rectangle. In the M lane, an additional isoform is disclosed (the M lane is boxed in yellow). On the right are the schematic representations of both the normal and aberrant isoforms, with exon 1 and 2 in blue and the pseudo-exon (Ψ) in orange. (**c**) Sanger sequencing electropherograms of reference (right panel) and abnormal PCR products (left panel) showing the splicing junction between exon 1 and exon 2, and between exon 1 and the pseudo-exon, respectively.

## Data Availability

Data supporting the findings of this study are available within the main text of the article and in the Appendix A. Additional data are available from the corresponding author on reasonable request.

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
