# Peer review of "Deep Intronic ETFDH Variants Represent a Recurrent Pathogenic Event in Multiple Acyl-CoA Dehydrogenase Deficiency"

_ijms, 2024, doi:10.3390/ijms25179637_

Round 1
Reviewer 1 Report
Comments and Suggestions for Authors
Summary: The authors aimed to give a brief overview of the phenotypic spectrum in multiple acyl-CoA dehydrogenase deficiency and highlight the current genetic landscape. Utilising a medical case as an example, the authors demonstrated that clinical exome sequencing may not provide a definitive molecular genetic diagnosis and that more comprehensive genetic analysis (whole genome sequencing) may be needed. The authors identified a deep-intronic variant identified in ETFDH and the manuscript focused heavily on the functional validation required to determine pathogenicity of this intronic variant.
General comments: Your methods section is very comprehensive and would assist other groups in replicating your analytical pipeline, if they wished to do so. Additionally, your figures and legends are well planned and written.
Author list: It is difficult to determine if any of the authors in the authorship are scientists. Given that your manuscript focuses heavily on the functional validation of the identified variants (as well as the described bioinformatic work), I would be very surprised if there were no scientists included in the manuscript. If there were no scientists included in the authorship, is this an oversight?
Line 8: You appear to be missing a senior author (name and *).
Lines 78-83: I think it's important to highlight that newborn screening is a screening tool (and also discuss the current dried blood spot acylcarnitine analytes being used around the world to trigger the algorithm for follow-up). I also think that it's important to highlight that when a patient screens positive on newborn screening, diagnostic biochemical genetic testing in plasma (acylcarnitine) and urine (organic acids) is typically performed prior to confirmatory molecular genetic testing.
Lines 92-95: Citation is needed
Lines 118: Add normal reference ranges
Lines 122-123: Add the abnormal levels with reference ranges (or at least the number of time the upper limit of normal for each analyte)
Lines 123-125: Was the organic acid analysis qualitative or quantitative? Was it performed by GC/MS?
Lines 127-130: Add reference ranges
Section 2.1 Patient: Did she clinically present prior to NBS results? Any hypotonia or hepatomegaly? Was an MRI of the brain performed? Any evidence of hypoglycemia? Were riboflavin levels measured?
Lines 204-206: With the extra "and", I find this sentence to be a little confusing.
Line 211: You should cite the actual publication for Splice AI. Jaganathan et al. 2019.
Lines 214-224: Add NM transcript for the intronic variant in the text
Line 239: Define ESE
Line 241-242: Define SF2/ASA ESE, etc
Line 244: Add the three variants in brackets to provide additional clarity
Line 295: Add the word "molecular" to definitive diagnosis - "definitive molecular diagnosis"
Lines 315-317: Add the citation for the other publications describing the variants identified in this intronic region.
Lines 346-349: I would reconsider the phrasing of this concluding statement as you and focussing on therapeutic intervention. Your patient passed away very early in life (before the possibility of splicing therapeutics to be explored).
In your methods section, consider integrating PDIVAS into your analytical pipeline (Kurosawa, R., Iida, K., Ajiro, M. et al. PDIVAS: Pathogenicity predictor for Deep-Intronic Variants causing Aberrant Splicing. BMC Genomics 24, 601 (2023). https://doi.org/10.1186/s12864-023-09645-2). The algorithm greatly reduces the number of false positives.
In your conclusion section, you should consider reclassifying both variants in the proband (and listing the ACMG criteria utilised given that you have confirmed phase and have also provided functional evidence).
References section: I'm having trouble identifying your referencing style. There appears to be missing information for reference numbers 3 and 19. Were these webpages? Additionally, there is a random number 48 towards the bottom of the reference list.
Just as a general comment, please ensure that your variants are submitted to ClinVar.
Comments on the Quality of English LanguageYou need to ensure that all gene names are italicised. Additionally, words such as in silico etc, should also be italicised. You may wish to consider utilising an editing service to ensure syntax and grammar is correct throughout the manuscript and to also ensure that the words that need to be italicised are.
Author Response
General comments: Your methods section is very comprehensive and would assist other groups in replicating your analytical pipeline, if they wished to do so. Additionally, your figures and legends are well planned and written.
Response 1 : We are grateful for this consideration.
Author list: It is difficult to determine if any of the authors in the authorship are scientists. Given that your manuscript focuses heavily on the functional validation of the identified variants (as well as the described bioinformatic work), I would be very surprised if there were no scientists included in the manuscript. If there were no scientists included in the authorship, is this an oversight?
Response 2 : We confirm that the authors are all scientist. Orcid ID of authors was added during the submission. Anyhow we are available to furnish other references if required.
Line 8: You appear to be missing a senior author (name and *).
Response 3: Thank you for pointing this out. We assure that no senior author was missing, the and was cancelled in the last version of the manuscript.
Lines 78-83: I think it's important to highlight that newborn screening is a screening tool (and also discuss the current dried blood spot acylcarnitine analytes being used around the world to trigger the algorithm for follow-up). I also think that it's important to highlight that when a patient screens positive on newborn screening, diagnostic biochemical genetic testing in plasma (acylcarnitine) and urine (organic acids) is typically performed prior to confirmatory molecular genetic testing.
Response 4 : We agree with this comment. Therefore, the topic is discussed on lines 72-77
Lines 92-95: Citation is needed
Response 5: We have accordingly added the citation.
Lines 118: Add normal reference ranges
Lines 122-123: Add the abnormal levels with reference ranges (or at least the number of time the upper limit of normal for each analyte).
Response 6: We agree with this consideration. All reference ranges and abnormal levels were added in the latest version of the manuscript. This will make clinical description more detailed and rigorous.
Lines 123-125: Was the organic acid analysis qualitative or quantitative? Was it performed by GC/MS?
Response 7: Thank you for this observation. We added in the latest version of manuscript that both quantitative and qualitative analysis were performed by GC/MS.
Lines 127-130: Add reference ranges
Response 8 : We agree with this consideration. All reference ranges and abnormal levels were added in the latest version of the manuscript. This will make clinical description more detailed and rigorous.
Section 2.1 Patient: Did she clinically present prior to NBS results? Any hypotonia or hepatomegaly? Was an MRI of the brain performed? Any evidence of hypoglycemia? Were riboflavin levels measured?
Response 9:
She was significantly hypotonic, no hepatomegaly and no hypoglycemia. MRI was not performed because very unstable condition of the patient.
Riboflavin levels were not measured.
Lines 204-206: With the extra "and", I find this sentence to be a little confusing.
Response 10: Thank you for this consideration. The sentence was rephrased in order to be clearer.
Line 211: You should cite the actual publication for Splice AI. Jaganathan et al. 2019.
Response 11: We have accordingly added the citation.
Lines 214-224: Add NM transcript for the intronic variant in the text.
Response 12: We have accordingly added NM transcript
Line 239: Define ESE
Line 241-242: Define SF2/ASA ESE, etc
Response 13: Thank you for pointing this out. We accordingly defined all acronyms.
Line 244: Add the three variants in brackets to provide additional clarity
Response 14: We agree with this comment. The variants were added in brackets in the latest version of the manuscript.
Line 295: Add the word "molecular" to definitive diagnosis - "definitive molecular diagnosis".
Response 15: we have accordingly added he word molecular.
Lines 315-317: Add the citation for the other publications describing the variants identified in this intronic region.
Response 16: Thank you for pointing this out. The citation “Nogueira C, Silva L, Marcão A, Sousa C, Fonseca H, Rocha H, et al. Role of RNA in Molecular Diagnosis of MADD Patients. Biomedicines. 2021 May;9(5) “was added.
Lines 346-349: I would reconsider the phrasing of this concluding statement as you and focussing on therapeutic intervention. Your patient passed away very early in life (before the possibility of splicing therapeutics to be explored).
Response 17: Thank you for pointing this out. The sentences were rephrased according to your indication, emphasizing the therapeutic aspect.
In your methods section, consider integrating PDIVAS into your analytical pipeline (Kurosawa, R., Iida, K., Ajiro, M. et al. PDIVAS: Pathogenicity predictor for Deep-Intronic Variants causing Aberrant Splicing. BMC Genomics 24, 601 (2023). https://doi.org/10.1186/s12864-023-09645-2). The algorithm greatly reduces the number of false positives.
Response 18: We thank you for highlighting the PDIVAS tool which we now cite in the revised version of the manuscript.
In your conclusion section, you should consider reclassifying both variants in the proband (and listing the ACMG criteria utilized given that you have confirmed phase and have also provided functional evidence).
Response 19: Thank you for pointing this out. The ACMG criteria utilized for the classification of the variant c.1852T>C are already described in the result section. We added ACMG criteria utilized for intronic variant on the discussion on the revised version of the manuscript.
References section: I'm having trouble identifying your referencing style. There appears to be missing information for reference numbers 3 and 19. Were these webpages? Additionally, there is a random number 48 towards the bottom of the reference list.
Response 20: Thank you for pointing this out. We modified the references as requested. Number 48 makes part of reference n 26, maybe there was a little pagination mistake.
Just as a general comment, please ensure that your variants are submitted to ClinVar.
Response 21: We’re proceeding with the variant’s submission on ClinVar
Comments on the Quality of English Language
You need to ensure that all gene names are italicised. Additionally, words such as in silico etc, should also be italicised. You may wish to consider utilising an editing service to ensure syntax and grammar is correct throughout the manuscript and to also ensure that the words that need to be italicised are.

Reviewer 2 Report
Comments and Suggestions for Authors
Strengths of the Study
Comprehensive Approach: The study employs a combination of whole genome sequencing (WGS) and RNA sequencing (RNAseq) to achieve a definitive molecular diagnosis in a newborn with Multiple Acyl-CoA Dehydrogenase Deficiency (MADD). This approach is highly commendable as it addresses the limitations of standard gene panel or exome sequencing, which often fail to detect deep intronic mutations that can affect splicing.
Identification of Novel Mutation: The discovery of a unique deep intronic mutation in the ETFDH gene (c.35-959A>G) is significant. The study adds to the existing knowledge by identifying a third mutation in intron 1 of the ETFDH gene, highlighting a possible region of vulnerability that may have broader implications for other genes.
Importance of RNA Sequencing: The study underscores the utility of RNA analysis in genome-wide testing. By incorporating RNA sequencing, the authors were able to reveal the functional consequences of intronic mutations, which is a critical step for accurate genetic diagnosis and better understanding of the disease pathology.
Detailed Methodology: The paper provides a comprehensive description of the methods used, including the bioinformatics tools and databases for variant analysis. This transparency allows for reproducibility and further validation by other researchers.
Weaknesses of the Study
Lack of Broader Context: While the study makes a significant discovery, it lacks discussion on how the findings can be generalized to other populations or conditions. The paper would benefit from a broader context regarding the prevalence and implications of deep intronic mutations in other genetic disorders.
Limited Sample Size: The study focuses on a single case, which limits the ability to generalize the findings. While this is typical for rare diseases, the paper could have strengthened its conclusions by including data from additional cases or families to demonstrate the consistency and recurrence of similar intronic mutations.
Insufficient Functional Validation: Although in silico predictions and some functional assays were used to validate the impact of the mutation, the study could benefit from additional experimental validation. For example, demonstrating the functional impact of the identified mutation in cellular or animal models would provide more robust evidence of its pathogenicity.
Clinical Implications and Future Directions: The paper briefly mentions the importance of RNA-based investigations for genetic diagnosis and counseling but does not elaborate on how these findings could influence clinical practice. More discussion on the potential for developing new diagnostic guidelines or personalized treatment strategies would enhance the study's relevance to clinical applications.
Recommendations
Expand the Study to Include More Cases: Future studies should aim to include a larger cohort of patients with suspected MADD or similar metabolic disorders. This would help validate the findings of deep intronic mutations and their impact on disease presentation.
Functional Studies: To strengthen the evidence, the authors should consider conducting more extensive functional studies. For instance, creating a cellular model with the identified mutation to observe its effects on ETFDH expression and function could provide more insight into the mutation's pathogenic mechanisms.
Broader Implications: The authors should explore the potential implications of their findings for other genetic disorders. Given the study’s suggestion that intronic mutations may be more common than currently believed, further research into the prevalence and impact of such mutations across different genes and conditions is warranted.
Conclusion
This study makes a valuable contribution to the field of genetic diagnostics by highlighting the importance of combining whole genome and RNA sequencing to uncover deep intronic mutations that affect splicing. However, the limited sample size and lack of extensive functional validation suggest that further research is needed to fully understand the clinical implications of these findings
Comments on the Quality of English LanguageMinor changes in English language must be done.
Author Response
Comments and Suggestions for Authors
Strengths of the Study
Comprehensive Approach: The study employs a combination of whole genome sequencing (WGS) and RNA sequencing (RNAseq) to achieve a definitive molecular diagnosis in a newborn with Multiple Acyl-CoA Dehydrogenase Deficiency (MADD). This approach is highly commendable as it addresses the limitations of standard gene panel or exome sequencing, which often fail to detect deep intronic mutations that can affect splicing.
Identification of Novel Mutation: The discovery of a unique deep intronic mutation in the ETFDH gene (c.35-959A>G) is significant. The study adds to the existing knowledge by identifying a third mutation in intron 1 of the ETFDH gene, highlighting a possible region of vulnerability that may have broader implications for other genes.
Importance of RNA Sequencing: The study underscores the utility of RNA analysis in genome-wide testing. By incorporating RNA sequencing, the authors were able to reveal the functional consequences of intronic mutations, which is a critical step for accurate genetic diagnosis and better understanding of the disease pathology.
Detailed Methodology: The paper provides a comprehensive description of the methods used, including the bioinformatics tools and databases for variant analysis. This transparency allows for reproducibility and further validation by other researchers.
Weaknesses of the Study
Lack of Broader Context: While the study makes a significant discovery, it lacks discussion on how the findings can be generalized to other populations or conditions. The paper would benefit from a broader context regarding the prevalence and implications of deep intronic mutations in other genetic disorders.
Limited Sample Size: The study focuses on a single case, which limits the ability to generalize the findings. While this is typical for rare diseases, the paper could have strengthened its conclusions by including data from additional cases or families to demonstrate the consistency and recurrence of similar intronic mutations.
Insufficient Functional Validation: Although in silico predictions and some functional assays were used to validate the impact of the mutation, the study could benefit from additional experimental validation. For example, demonstrating the functional impact of the identified mutation in cellular or animal models would provide more robust evidence of its pathogenicity.
Clinical Implications and Future Directions: The paper briefly mentions the importance of RNA-based investigations for genetic diagnosis and counseling but does not elaborate on how these findings could influence clinical practice. More discussion on the potential for developing new diagnostic guidelines or personalized treatment strategies would enhance the study's relevance to clinical applications.
Recommendations
Expand the Study to Include More Cases: Future studies should aim to include a larger cohort of patients with suspected MADD or similar metabolic disorders. This would help validate the findings of deep intronic mutations and their impact on disease presentation.
Functional Studies: To strengthen the evidence, the authors should consider conducting more extensive functional studies. For instance, creating a cellular model with the identified mutation to observe its effects on ETFDH expression and function could provide more insight into the mutation's pathogenic mechanisms.
Broader Implications: The authors should explore the potential implications of their findings for other genetic disorders. Given the study’s suggestion that intronic mutations may be more common than currently believed, further research into the prevalence and impact of such mutations across different genes and conditions is warranted.
Conclusion
This study makes a valuable contribution to the field of genetic diagnostics by highlighting the importance of combining whole genome and RNA sequencing to uncover deep intronic mutations that affect splicing. However, the limited sample size and lack of extensive functional validation suggest that further research is needed to fully understand the clinical implications of these findings
Comments on the Quality of English Language
Minor changes in English language must be done.
REPLY
We sincerely appreciate your thoughtful and detailed comments on our manuscript. Your suggestions are indeed valuable, and we agree that they will be noteworthy in similar cases, particularly when time and resources allow for the inclusion of experiments with cellular and animal models. We understand the importance of exploring the broader implications of deep intronic mutations and their contribution to the diagnostic challenges in rare diseases. In the discussion section of our manuscript, we have tried to contextualize our findings by referencing other known cases of deep intronic mutations and discussing their related implications for reaching a definitive diagnosis. We hope this provides a solid base for understanding the potential significance of such mutations in our case. Moreover, we recognize the need to expand the case history of MADD (Multiple Acyl-CoA Dehydrogenase Deficiency) cases. It is indeed in our future plans to outline diagnostic guidelines, particularly for suspected cases where only a single mutation is identified. We believe that expanding this case history will greatly contribute to the development of more robust diagnostic criteria, thereby benefiting the broader medical community and improving patient outcomes. Lastly, we fully agree with your observation that more efforts are required to understand the global impact of deep intronic mutations in rare diseases, as well as their real prevalence, which we believe is probably underestimated.
As WGS approaches for genetic diagnosis of diseases become more pervasive and widespread, it will be necessary to address the vast number of intronic variants whose roles are not yet clear. Consequently, the need for functional validation methods will become a primary concern. We are committed to contributing to this important area of research and look forward to further investigations that will deepen our understanding of these mutations' roles.
We thank you for your constructive feedback, for your time and thoughtful comments.
